# STINGing Cancer: Development, Clinical Application, and Targeted Delivery of STING Agonists

**DOI:** 10.3390/ijms26189008

**Published:** 2025-09-16

**Authors:** Yannick Gabriel Nerdinger, Amanda Katharina Binder, Franziska Bremm, Niklas Feuchter, Niels Schaft, Jan Dörrie

**Affiliations:** 1Department of Dermatology, Universitätsklinikum Erlangen, Friedrich-Alexander-Universität Erlangen-Nürnberg, 91054 Erlangen, Germany; yannick.nerdinger@fau.de (Y.G.N.); amanda.binder@uk-erlangen.de (A.K.B.); franziska.bremm@uk-erlangen.de (F.B.); niklas.feuchter@uk-erlangen.de (N.F.); niels.schaft@uk-erlangen.de (N.S.); 2Comprehensive Cancer Center Erlangen European Metropolitan Area of Nuremberg (CCC ER-EMN), 91054 Erlangen, Germany; 3Deutsches Zentrum Immuntherapie (DZI), 91054 Erlangen, Germany; 4Bavarian Cancer Research Center (BZKF), 91052 Erlangen, Germany; 5Comprehensive Cancer Center Würzburg Erlangen Regensburg Augsburg (CCC-WERA), 91054 Erlangen, Germany

**Keywords:** STING agonist, small molecules, nano therapy, immunotherapy, cancer therapy, tumor microenvironment, tumor-associated macrophages

## Abstract

As cancer incidence continues to rise and conventional therapies remain of limited effectiveness, the search for novel and innovative cancer treatments is ongoing. In recent years, immunotherapies, including checkpoint inhibitors and cell-based approaches such as CAR-T cell therapy, have revolutionized the treatment of cancer. However, response rates even to well-established immunotherapies remain low in several types of cancer. Therefore, various novel immunomodulatory substances are currently under investigation, among them agonists of the intracellular signaling protein STING (STimulator of INterferon Genes). Activation of the STING signaling pathway can alter the cytokine profile within the tumor microenvironment (TME) and reshape the function of various immune cells. STING agonists have yielded promising results in preclinical studies, but this success has not yet been replicated in clinical trials. Consequently, STING agonists are optimized for greater potency and combined with nanotechnologies to enhance biodistribution and achieve sustained accumulation within the TME. This review summarizes a selection of STING agonists evaluated in clinical trials to date and discusses their effects on tumor-infiltration immune cells, especially macrophages. It highlights emerging candidates currently under investigation in preclinical studies, and explores nanotechnological approaches for their combinational use to enhance therapeutic efficacy.

## 1. Introduction

Despite major advances in cancer immunotherapy through checkpoint inhibitors and CAR-T cell therapies in recent years, substantial challenges remain. Genetically engineered CAR-T cells revolutionized cancer therapy for blood cancer, but proved generally ineffective in solid tumors, primarily because access to the tumor tissue constitutes a major bottleneck for these cellular therapies. Furthermore, the immunosuppressive tumor microenvironment impairs the anti-tumoral function of both endogenous and genetically engineered immune cells. To address solid tumors, strategies involve the development of subcellular therapies that specifically target immune cells already residing within the tumor microenvironment (TME). Immune checkpoint inhibitors (ICI) that block inhibitory receptors on tumor-specific T cells require not only an immunogenic epitope repertoire on the cancer cells, but also T-cell infiltration into the tumor. Given the critical role of the immune cell composition within the TME in determining the therapeutic outcome of immunotherapies, the polarization of innate immune cells towards a pro-tumoral phenotype is promising strategy. Here, macrophages are of particular interest, as they are abundant within the TME and can exhibit both pro- and anti-tumoral polarization states, which remain plastic and reversible. One major driver of macrophage polarization is the STING (STimulator of INterferon Genes) signaling pathway. It has been shown that STING signaling in macrophages diminishes their suppressive functions and enhances their contribution to tumor clearance through pro-inflammatory effector activity [1]. This pro-inflammatory shift extends across multiple immune cell subsets within the TME, ultimately reinforcing immune cell cytotoxicity. This review provides an overview of STING signaling in immune cells, especially macrophages, and examines preclinical and clinical data on STING agonists, with a particular focus on their role in STING-mediated macrophage reprogramming within the TME.

## 2. Macrophage Involvement in Tumor Elimination

Since Mills et al. postulated the macrophage polarization dichotomy in 2000, macrophages have been broadly categorized into pro-inflammatory M1 and anti-inflammatory M2-LIKE states [2]. Under in vitro culture conditions, this M1/M2 paradigm appears straightforward, as polarization with IFNγ or IL-4 induces clear-cut antagonistic effects on macrophage polarization [3]. However, it is now widely accepted that macrophage polarization is shaped by a multitude of external signals, resulting in a continuum of activation states [3]. In this spectrum, the classically defined in vitro-induced M1 and M2 macrophages represent rather the extreme ends, with many intermediate states existing in between [4,5,6].

Despite this conceptual advancement, the nomenclature remains inconsistent. While many researchers continue to rely on the M1/M2 classification, others advocate for more precise designation at least of in vitro-generated macrophages based on their activating stimuli—for example, M(IL-4), M(LPS), or M(IFNγ) [3], or their function in their natural habitat, e.g., iron-cycling macrophages [7]. However, the main challenge in defining macrophage polarization lies in the absence of a single definitive marker, unlike many other cell lineages. Instead, the identification of macrophage activation states requires the analysis of multiple markers, which are differentially up- or downregulated depending on the stimulus [3,5,8,9]. As a result, both human in vitro and in vivo macrophage activation states still lack a universally accepted classification framework.

Taking all this into account, we use the terms *M1-like* and *M2-like* throughout this review. We are aware that this terminology is not ideal; however, the sources we reference and the data we present predominantly use the M1-/M2 nomenclature. Therefore, we have chosen to retain the original terminology as used in the cited literature, rather than introducing alternative terms.

The herein described M1-like macrophages can be induced by pathogens and antigen-experienced T cells as well as IFNγ, LPS, and GM-CSF. In their activated state, M1-like macrophages act cytotoxically and release reactive oxygen species (ROS) and nitric oxide (NO), as well as pro-inflammatory cytokines such as TNFα, IL-1β, IL-6, IL-12, and IL-18 [10,11,12,13]. Moreover, they efficiently eliminate and phagocytose infectious threats as well as malignant cells and establish an inflammatory environment that enhances anti-tumoral T_H_1 effector functions. In contrast, M2-like macrophages are polarized by other cytokines such as IL-4 and IL-13, and in turn secrete chemokines including CCL17, CCL22, CCL2, and CCL5 as well as anti-inflammatory mediators such as arginase 1, IL-10, and TGFβ. As result, a strong T_H_2 and regulatory T-cell (Tregs) response is induced [2,14,15], contributing to immune tolerance, resolution of inflammation, and tissue repair [16,17,18,19]. The presence of macrophages polarized towards an M2-like phenotype has been associated with reduced target elimination in both infections and cancer [20,21,22,23].

Within the TME, macrophages constitute approximately 5% to 50% of immune cells and are predominantly polarized towards the M2-like phenotype, commonly referred to as tumor-associated macrophages (TAMs) [24,25]. Various studies have confirmed the prevalence of M2-like macrophages in treatment-resistant tumors, while inflammatory M1-like macrophages appear to be underrepresented or functionally suppressed within the TME [26,27,28,29].

Since macrophage polarization depends on complex signaling programs, triggered by distinct stimuli and mediators, and research into macrophage plasticity still expands, it is becoming increasingly evident that macrophage differentiation is not terminal. Instead, additional stimulation can induce repolarization from one phenotype to another. Therefore, therapeutics that shift M2-like to M1-like macrophages may be a crucial factor in effective cancer elimination [30].

## 3. STING Signaling Determines Macrophage Polarization

One important mediator of macrophage polarization is the endoplasmic reticulum (ER)-associated signaling molecule STING, which is a sensor of cytosolic double stranded DNA (dsDNA) [1,31]. Figure 1 provides an overview of the STING signaling pathway in macrophages and its cellular effects. cGAS, the cyclic GMP-AMP (cGAMP) synthase, detects cytosolic dsDNA and forms a complex, which induces dimerization and subsequent activation of cGAS [32,33]. The recognized dsDNA can originate from either self or foreign sources, enabling the cell to react to invading pathogens as well as to disruptions in nuclear or mitochondrial membrane integrity. Upon activation, cGAS synthesizes the second messenger cGAMP [34,35] that binds to the protein STING, which spans the membrane of the ER [34,36,37]. This leads to dimerization and activation of STING [38]. Once activated, STING translocates to ER-Golgi intermediate compartments recruiting the TANK binding kinase 1 (TBK1) [39]. TBK1 undergoes autophosphorylation and subsequently phosphorylates STING [40], transforming it into a scaffold that facilitates the recruitment of interferon regulatory factor 3 (IRF3) [41]. After phosphorylation by TBK1, IRF3 dimerizes and translocates into the nucleus, where it initiates the transcription of type 1 interferons (IFN-1) and pro-inflammatory cytokines [42].

Apart from the canonical IRF3 activation, STING can also induce activation of the transcription factor nuclear factor kappa B (NF-κB) through the IκB kinase (IKK), thus further enhancing the type 1 interferon response [43]. This non-canonical pathway additionally induces the production of pro-inflammatory cytokines, including IL-6 and IL-12.

Furthermore, STING signaling has been linked to the activation of the NLRP3 (NOD-like receptor family pyrin domain-containing-3) and AIM2 (absent in melanoma 2) inflammasomes [44,45,46], resulting in the secretion of the activated cytokines IL-18 and IL-1β [46,47]. Extensive inflammasome activation can also induce programmed cell death mechanisms such as apoptosis, pyroptosis, or necrosis in affected cells [48,49].

In conclusion, the STING signaling pathway initiates a unique transcriptional program in macrophages including the activation of NF-κB and IRF3, which lead to the production of pro-inflammatory cytokines. Hence, the STING pathway in macrophages is primarily associated with the polarization toward an M1-like phenotype. Indeed, intratumoral injection of cGAMP facilitates the recruitment and activation of classically activated macrophages in a STING dependent manner [50], while a loss in STING signaling was associated with increased susceptibility to specific pathogens due to impaired IFN-1 production [38,51,52].

## 4. Categories of STING Agonists

The STING signaling pathway plays a crucial role in promoting a pro-inflammatory effector phenotype, not only in macrophages but also in various other immune cell subsets. This immune reprogramming has been shown to drive effective anti-tumor responses, contributing to tumor elimination. Building on these findings, numerous STING agonists have been developed to synthetically activate STING and induce a robust pro-inflammatory response (Figure 2). These agonists are being explored as safe and effective alternatives to conventional cancer therapies, particularly for tumors resistant to standard treatment approaches [53,54].

In general, STING agonists can be divided into different subgroups including synthetic cyclic dinucleotides (CDNs), non-cyclic dinucleotides (non-CDNs), antibody drug conjugates (ADCs), and indirect STING agonists. Synthetic CDN STING agonists mimic natural CDN compounds like cGAMP and their engagement with STING. They were the first STING agonists to enter clinical investigation, with promising initial success. However, most CDNs are rapidly metabolized upon administration, leading to a reduced stability and a short serum half-life [55,56,57]. In contrast, non-cyclic dinucleotide agonists (non-CDNs) were designed for an increased stability and solubility to enhance STING activation while reducing off-target effects. To enhance the potential of STING activation within the TME, STING agonists can be employed as part of ADCs, nanocarriers, or bacterial vectors to enable targeted delivery to cells within the TME. This controlled activation of STING reduces off-target effects and prevents systemic inflammation. Furthermore, these systems shield the agonist from degradation and ensure prolonged STING activation [58,59,60,61,62,63,64]. In contrast to the previously described classes, some substances function as indirect STING agonists. These agonists do not directly engage with STING, but are designed to increase signaling either downstream or upstream of STING activation. Targets of indirect STING agonists usually include the three prime repair exonuclease (TREX1), ectonucleotide pyrophosphatase/phosphodiesterase family member 1 (ENPP1), cGAS, and TBK1 [65,66,67,68,69,70,71]. The following part summarizes different STING agonists and their explicit effect on the immune cell composition as well as soluble factors within the TME.

### 4.1. STING Agonists in Clinical Trials for Cancer Therapy and Their Immunological Effects

#### 4.1.1. IMSA101

A comprehensive summary of clinical investigations involving STING agonists is presented in Table 1. The first agonist to be reviewed here is the synthetic CDN analog IMSA101, also named GB492. Preclinical studies on mice examined the efficacy of intratumorally administered IMSA101 in combination with CAR-T cell treatment. These studies showed a significantly higher release of pro-inflammatory cytokines (e.g., IL-18) and a downregulation of TGFβ and IL-10 as well as necrosis within the tumors of mice receiving the combination therapy. The treatment elevated levels of M1-like macrophages and NK cells in the tumor, while M2-like macrophage levels decreased. Interestingly, no significant difference in total M2-like macrophage numbers was found, suggesting an increased infiltration of M1-like macrophages, rather than a reprogramming of M2-like macrophages. These macrophages exhibited an elevated IL-18 signature, further stimulating immune cell activation and enhancing the anti-tumor response. Moreover, significant infiltration of M1-like macrophages was observed with IMSA101 monotherapy and combination therapy with anti-PD-L1 compared to anti-PD-L1 treatment alone [72]. The impact of STING agonists on immune cell populations within the TME is summarized in Table 2; their effects on soluble factors are shown in Table 3.

IMSA101 then progressed to phase 1 clinical evaluation as single agent and in combination with checkpoint inhibitors (NCT4020185). Tumor shrinkage was observed in all settings; however, a partial response to treatment was only seen in one of the 13 patients within the combinational cohort, none within the monotherapy cohort [73]. Based on these findings, two phase 2 trials for IMSA101 in combination with checkpoint inhibitors and personalized ultra-fractionated stereotactic adaptive radiotherapy (PULSAR) were initiated in 2023 (NCT05846646, NCT05846659) [74]. The trials were discontinued in 2024 due to the sponsor’s decision. Patients who showed a favorable response to treatment with IMSA101 were positioned in a roll-over study to continue receiving the treatment (NCT06026254). However, a third clinical trial investigating IMSA101 in combination with PULSAR for renal cell carcinomas was started in April 2025 (NCT06601296).

#### 4.1.2. ADU-S100 (MIW 815)

Another synthetic CDN agonist currently in clinical research is ADU-S100 (also referred to as MIW 815 or ML-RR-S2-CDA). ADU-S100 was optimized for an increased affinity to human STING (hSTING) and higher activation efficacy [55]. In murine cancer models, intratumoral injection of ADU-S100 promoted the release of pro-inflammatory cytokines and enhanced infiltration of immune cells such as macrophages, CD8^+^ T cells, NK cells, and neutrophils into the TME, leading to a robust anti-tumor response while decreasing mortality [55,75,76]. In this study, TME-associated macrophages were identified as key producers of IFN-1 [55].

In mouse models of peritoneal carcinomatosis of colon cancer (PCCC), intraperitoneal injection of ADU-S100 was able to reprogram the suppressive TME into a highly inflammatory milieu. STING signaling in macrophages led to the upregulation of M1-associated markers on the cell surface, reprogramming them into a pro-inflammatory state. Immune remodeling in the TME has been linked to IFN-1 signaling as well as CD8^+^ T-cell activity. Treated mice upregulated inhibitory PD-L1 molecules on intratumoral and myeloid cells. Correspondingly, the combination of ADU-S100 with ICIs further enhanced anti-tumor responses as well as M1-like macrophage recruitment, while the proportion of M2-like macrophages and myeloid derived suppressor cells decreased [76].

Two phase 1 clinical trials were initiated in 2016 to assess the therapeutic potential of ADU-S100 as monotherapy or in combination with checkpoint inhibitors (NCT02675439, NCT03172936). A phase 2 clinical trial for ADU-S100 started in 2019 for patients with head and neck cancers. Administration of the agonist was tolerated well in both mono- and combination therapy, while additionally displaying limited serum half lifetime. Despite the promising functionality of ADU-S100 in preclinical evaluations, the studies conducted on patients indicated only limited clinical activity of the agonist. Tumor growth inhibition and indications of systemic immune activation were observed in only a subset of patients, and the therapy had no effect on macrophage recruitment [57,77]. All trials involving ADU-S100 were eventually terminated due to the lack of substantial anti-tumor response. Current research is focused on developing nanocarriers encapsulating ADU-S100 to enhance its resistance to degradation [78].

#### 4.1.3. SB 11285

SB 11285 is a synthetic CDN STING agonist, which can be administered not only intratumorally, but also systemically via intraperitoneal and intravenous routes [72]. This may promote the recruitment of effector cells from other tissues and the bloodstream towards the tumor site.

Preclinical studies in mice confirmed STING activation in several immune cells, including monocytes. SB 11285 demonstrated a favorable safety profile while inducing strong production of IFN-1, as well as other cytokines, chemokines, and interferon stimulated genes (ISGs). Importantly, inflammation remained localized within the TME not only after intratumoral injection [79], but also after systemic application [80]. Direct exposure of the agonist to tumor tissue also induced apoptosis of target cells in a STING-dependent manner. Observed anti-tumor effects included the infiltration of macrophages, CD8^+^ T cells, and NK cells, while the infiltration of CD4^+^ T cells and Tregs was reduced. Additionally, combining SB 11285 with immune checkpoint blockade therapy (anti-PD-1 or anti-CTLA-4) further enhanced anti-tumor efficacy and immune activation [81].

In vitro studies confirmed the binding of SB 11285 to different human STING variants. Exposing human PBMCs, monocytes, and monocyte-derived dendritic cells (DCs) to SB 11285 resulted in the production of IFN-1, as well as other cytokines and chemokines [82].

A phase 1a/b clinical trial evaluating SB 11285, either alone or in combination with the PD-L1 inhibitor Atezolizumab, in patients with advanced solid tumors completed its evaluation in July 2024 but the results were not yet published (NCT04096638).

#### 4.1.4. TAK-676 (Dazostinag)

TAK-676 (also known as Dazostinag) is another synthetic CDN STING agonist currently under clinical investigation. It has been optimized to achieve enhanced serum stability and tissue permeability, enabling intravenous administration for systemic delivery [83,84,85].

Preclinical data confirmed the binding of TAK-676 to both human and murine STING. Enhanced recruitment of immune cells was observed in tumor tissue, as well as an increased abundance of NK cells and DCs in the lymph nodes. Treatment led to a robust anti-tumor response, characterized by pro-inflammatory mediators such as IFNα, IFNγ, CXCL10, TNFα, and IL-6 [83]. Strong responses have been examined in the combination with checkpoint inhibitors or radiation therapy, with the potential to overcome treatment resistance [84,85]. An early phase 1 clinical trial assessed efficacy of micro-dosing TAK-676 alone and in combination with Carboplatin, Paclitaxel, or 5-FU (NCT06062602). The study confirmed rapid IFNγ production in multiple cell types, as well as the release of pro-inflammatory cytokines and chemokines in patients up to 24 h after micro-dosing. An increase in M1-like macrophages, CD8^+^ T cells, and NK cells was observed, while general safety was maintained [86]. Phase 1/2 clinical trials are ongoing to investigate dose optimization and efficacy of TAK-676 as monotherapy and in combination with immunotherapy and radiation therapy (NCT04420884, NCT04879849).

#### 4.1.5. E7766

The novel STING agonist E7766 is based on a CDN but contains an additional molecular bridge between the purine-residues. Designed for enhanced stability, it can be administered both intratumorally and intravenously. High-affinity binding to murine and human STING across several genotypes has been confirmed in in vitro binding models [87,88]. The agonist elicits robust activation of STING, exceeding the response observed in tests conducted with natural c-GAMP and CDNs [87,88,89,90,91]. In vivo analysis in mice showed that the administration of E7766 produced a highly effective anti-tumor response, characterized by infiltrating granulocytes, DCs, B cells, NK cells, and T cells, along with the production of IFNβ and CXCL10 (IP-10) [87,88,90]. Moreover, E7766 enhanced survival and tumor clearance in treatment resistant murine models [87].

Another preclinical study utilized an ADC, linking E7766 to an antibody against prostate-specific membrane antigen (PSMA) to target PSMA-expressing prostate cancer cells, delivering the drug E7766 with high specificity to the tumor. The agonist was found to accumulate in high abundance in tumor tissues compared to lower levels in plasma. The treatment induced a robust anti-tumor response, as assessed by the production of pro-inflammatory cytokines and tumor shrinkage. STING pathway activity was confirmed by the production of IFNβ, CXCL10, TNFα, and IL-6. Furthermore, reprogramming of M2-like macrophages into the pro-inflammatory M1-like subtype was observed in the TME [92].

Two phase 1 trials were initiated to investigate the clinical potential of E7766 as monotherapy in patients with solid tumors in February 2020 (NCT04144140, NCT04109092). One observed a systemic increase in the pro-inflammatory cytokines IFNα, IFNβ, IFNγ, TNFα, IL-6, and CXCL10 in treated patients, which could not be brought in correlation with the administered dose of E7766. Lower, rather than high doses of the agonist lead to an increase in CD8^+^ effector T cells [93]. Both trials were discontinued due to treatment-unrelated reasons.

#### 4.1.6. SYNB 1891

The bacterial vector STING agonist SYNB 1891 is a live strain of probiotic *E. coli Nissle*, modified to produce cyclic-di-AMP (CDA) under hypoxic conditions, which are often induced in solid tumors [94]. In co-cultures of SYNB 1891 with macrophages, the bacterial vector-derived STING agonist activated STING signaling and induced a dose-dependent expression of IFNβ in macrophages. This STING-dependent IFNβ production was shown to rely on the phagocytic uptake of the bacterial vector, highlighting the crucial role of phagocytes in SYNB 1891-induced anti-tumor responses [62].

Preclinical studies on mice confirmed that the vector proliferated within tumors, remained localized, and did not extravasate into the bloodstream. The production of CDA by the vector induced a dose-dependent increase in IFNβ, IFNγ, TNFα, IL-6, IL-1β, and GM-CSF expression. Apart from STING activation, SYNB 1891 also stimulated innate immune cells via pattern recognition receptors, further contributing to IFN-1 production. The agonist elicited a potent CD8^+^ T cell-dependent anti-tumor response, leading to tumor regression in treated mice [62]. A phase 1 clinical trial evaluated SYNB 1891 as monotherapy or in combination with Atezolizumab in patients with advanced or metastatic solid tumors or lymphomas (NCT04167137). The trial demonstrated initial efficacy, as indicated by the induction of ISGs, cytokines, chemokines, and T-cell response genes. However, it was ultimately terminated due to the sponsor’s decision. Within the 24 patients of the monotherapy cohort, four experienced cytokine release syndrome, and 54% discontinued treatment due to disease progression [61,95].

#### 4.1.7. ExoSTING (CDK-002)

The nanocarrier exoSTING (or CD-002) is an engineered exosome displaying high quantities of the glycoprotein PTGFRN, which enables specific targeting of antigen presenting cells (APCs) [96,97]. The exosomes are loaded with synthetic CDNs to activate STING in APCs after uptake [60]. Preclinical studies confirmed that this approach prevented the uncontrolled activation of immune cells residing within the TME, which reduces the risk of high inflammatory responses outside of tumor tissue. The agonist was retained at the site of injection and promoted strong cytokine responses without systemic exposure. The highest uptake of exoSTING was observed in DCs, monocytes, and M2-like macrophages. Interestingly, uptake and activation of M1-like macrophages remained low. Minor doses of exoSTING were sufficient to induce a high IFNβ signature in activated DCs and M2-like macrophages, along with enhanced infiltration of macrophages. The extent of IFNβ production was shown to correlate with the levels of macrophages localized within the tumor environment. This produced a pro-inflammatory shift within the TME, promoting high activation of T cells and reprogramming into T_H_1 effector subsets [60]. Lastly, treatment with exoSTING upregulated surface expression of PD-L1. Combination with PD-1 checkpoint blockade therapy has been shown to efficiently overcome this limitation [96]. Overall, intratumoral administration of exoSTING to tumor-bearing mice resulted in a robust and long-lasting anti-tumor response, which was highly dependent on TAM and CD8^+^ T cell effector mechanisms [60,96,97]. A phase 1/2 clinical trial of exoSTING in patients with advanced or metastatic, recurrent, injectable solid tumors was completed in 2022 (NCT04592484). The trial focused on tumors with a high abundance of the targeted APCs. The sponsor claimed that ExoSTING was effectively delivered to the tumor and retained within the tissue without systemic exposure to the bloodstream. Consequently, adverse events caused by systemic inflammation and cytokine exposure remained low. Tumor shrinkage was observed in injected tumors, as well as non-injected, distal tumors in a subset of patients. Low doses of exoSTING were able to induce substantial STING pathway activity in patients, along with the production of IFN-1 and CXCL10, indicating activation of the innate immune compartment. Eventually, treatment with exoSTING also induced activation of adaptive immune subsets, including effector CD8^+^ T cells. The migration of activated immune cells expressing ISGs from the tumor into the bloodstream was observed [98,99].

#### 4.1.8. ONM-501

Another STING agonist nanoparticle is ONM-501. The agonist uses the synthetic, pH-sensitive STING-activating polymer PC7A to encapsulate cGAMP. At physiological pH, the PC7A polymers form micelles, protecting the nanoparticle from degradation. Upon reaching the acidic TME, pH-induced dissociation of the micelle releases cGAMP from endocytosed nanoparticles, delivering the agonist cGAMP as well as the PC7A polymers to the target cells. Both cGAMP and PC7A polymers directly interact with STING and synergistically activate the protein. Additionally, the PC7A polymers stabilize STING molecules upon binding, slowing down their degradation and extending STING activity [100,101,102].

Preclinical studies confirmed dose-dependent activation of both mouse and human STING after intratumoral injection. ONM-501 was shown to accumulate within the tumor and induce rapid, prolonged STING signaling in the targeted cells. This signaling led to the expression of pro-inflammatory cytokines, including high levels of IFNβ and CXCL10. Notably, no cytokine storm due to systemic cytokine release was observed in treated animals. ONM-501 elicited a potent, CD8^+^ T cell-dependent anti-tumor response, inhibiting tumor growth in both injected and non-injected tumors. Moreover, combining ONM-501 with anti-PD-L1 immune checkpoint blockade enhanced anti-tumor responses in treated mice [102,103,104,105].

In preclinical studies involving the PC7A polymer alone, treatment resulted in an increase in M1-like macrophages, while the fraction of M2-like macrophages was reduced, suggesting a reprogramming of macrophages into a pro-inflammatory state [101].

ONM-501 is currently being evaluated in a phase 1 clinical trial in patients with solid tumors or lymphomas, either as a monotherapy or in combination with the anti-PD-1 antibody Cemiplimab (NCT06022029) [100].

### 4.2. STING Agonists and Their Effect on Macrophages in Preclinical Evaluations

Beside the STING agonists described above that have already been investigated in clinical trials, there are several additional candidates currently undergoing preclinical evaluation. These have demonstrated a great impact on pro-inflammatory macrophage polarization as well as promising anti-tumor effects in preclinical mouse models and are reviewed in more detail in the following section.

IACS-8803, also referred to as IMGS-203, is a CDN STING agonist, which effectively binds and activates both murine and human STING variants. Treatment with IACS-8803 induced tumor regression comparable to checkpoint inhibitor therapy and outperformed other CDN agonists like ADU-S100. IACS-8803 promoted increased infiltration of CD8^+^ T cells and NK cells into the TME and expanded the population of conventional DCs. It also induced functional repolarization of suppressive immune cells, surpassing the effects of ADU-S100 or cGAMP. Myeloid cells within the TME upregulated co-stimulatory molecules while downregulating suppressive markers. In human macrophages, IACS-8803 induced the downregulation of M2-associated markers such as CD163, TGFβ, and arginase, while promoting differentiation into pro-inflammatory M1-like macrophages expressing CD80, CD86, IL-6, and iNOS. Combination therapy with checkpoint inhibitors further enhanced CD8^+^ T-cell cytotoxicity, leading to complete tumor eradication in treated animals [106,107,108,109,110].

The non-CDN agonist MSA-2 was developed for systemic administration and can be delivered via subcutaneous, oral, or intratumoral injection. In murine cancer models, MSA-2 exhibited high permeability and preferentially targeted and activated STING within the acidic TME. It potently induced STING signaling, leading to the production of the pro-inflammatory cytokines IFNβ, TNFα, and IL-6. Importantly, MSA-2 treatment promoted a shift within the TME towards inflammation. Treated tumors showed an increased presence of M1-like macrophages, CD8^+^ T cells, and NK cells, correlating with elevated levels of CCL5, CXCL9, and CXCL10 chemokines. Immune cell subsets within the tumor tissue displayed a pro-inflammatory phenotype and increased cell–cell interactions [111,112].

Another non-CDN STING agonist, SR-717, has demonstrated great anti-tumor efficacy in preclinical studies. Mice treated with SR-717 exhibited a higher frequency of activated NK cells and CD8^+^ T cells in tumor tissue, lymph nodes, and the spleen. The agonist induced a dose-dependent production of IFNβ and upregulated PD-1 and PD-L1 expression. Combination therapy with ICIs enhanced tumor clearance compared to SR-717 monotherapy [113]. Additionally, Wang et al. report in a preprint that SR-717 polarizes M1-like macrophages from undifferentiated M0 macrophages in vitro [114]. A shift toward the predominance of pro-inflammatory M1-like macrophages in the tumor microenvironment was further demonstrated using SR-717 encapsulated in human serum albumin nanoparticles—a formulation optimized for more efficient and targeted delivery. This nanoparticle-based approach was more potent in inducing STING-mediated anti-tumor responses than administration of free SR-717 [115].

## 5. Conclusions: STING Agonists Drive Pro-Inflammatory Reprogramming in the TME

The reviewed STING agonists demonstrate the potential to induce STING-mediated immune responses for effective tumor clearance. By activating key immune cell populations within the tumor microenvironment, STING signaling helps to reshape the immune landscape toward a more pro-inflammatory and tumor-suppressive state [76,111].

STING signaling has been shown to enhance the activation and recruitment of NK cells, CD8^+^ T cells, DCs, and macrophages within the TME [72,76,79,83,87]. Both NK cells and CD8^+^ T cells exhibit increase—d cytotoxic effector functions [106,116], which are further reinforced by the upregulation of co-stimulatory molecules on macrophages and DCs [60,106,116]. TAMs display a higher expression of CD80, CD86, IL-6, iNOS, and TNFα, while the expression of CD163, TGFβ, and arginase is reduced—suggesting a shift toward a pro-inflammatory M1-like phenotype [76,106]. Although STING signaling has the potential to induce STAT6 expression and M2-like polarization [1,117], the reviewed data indicate a predominant polarization of macrophages into the M1-like state following STING agonist treatment [76,92,101]. Overall, STING activation is associated with a reduction in tumor-supportive immune cells, including myeloid derived suppressor cells, M2-like macrophages, and Tregs [72,80,118].

Clinical and preclinical studies consistently report a strong STING-dependent induction of pro-inflammatory mediators such as IFNβ, CXCL-10, TNFα, and IL-6, followed by the expression of ISGs [62,81,83,92]. This cytokine profile is characteristic of a robust type 1 immune response. Notably, macrophages, alongside CD8^+^ T cells, were identified as key producers of pro-inflammatory cytokines, including IFNβ and IL-18 [55,72,119]. Additionally, some STING agonists were shown to enhance IFNγ and GM-CSF secretion, further promoting the polarization and activation of M1-like macrophages [62,83].

Conversely, STING agonists also diminish the expression of immunosuppressive cytokines such as TGFβ and IL-10 [72]. Several studies indicate that selective targeting of immunosuppressive myeloid cells within the TME is sufficient to induce effective anti-tumor responses [60,120,121]. A shift toward the predominance of M1-like macrophages, coupled with a reduction in suppressive myeloid cells, creates an environment that enhances effector T-cell functions and strengthens anti-tumor immunity.

## 6. Limitations of STING as Therapeutic Target

With the promising results of early agonists, the cGAS-STING pathway has become a major focus in cancer immunotherapy. Numerous STING agonists are currently in development, with emerging clinical data supporting their potential. However, to date, none have been granted approval for clinical use by the FDA or the EMA due to insufficient treatment responses observed in clinical trials. Early trials of CDN STING agonists have demonstrated poor serum stability, limited permeability, and consequently, low efficacy in patients. Challenges for STING agonist are manifold. First, STING agonists show a poor cell internalization since most STING agonists are negatively charged which limits their permeability across the negatively charged cell membrane [122]. Secondly, they can elicit systemic inflammation, excessive cytokine release [123], and other severe adverse events, as reviewed in detail by Barber et al. [124] and Gehrcken et al. [125]. Although non-CDN STING agonists demonstrate improved serum stability and enhanced anti-tumor efficacy [112,126], the risk of treatment-related adverse events due to off-target effects remains a concern and limits the systemically applicable dose. Lastly, even if injected intratumorally to avoid systemic adverse effects, STING agonists fail to remain within the TME. They are rapidly cleared from the TME due to their low molecular weight, high water solubility, and high susceptibility to enzymatic degradation [57,122,127]. For a detailed discussion of the limitations and challenges of STING agonists we would like to point to a review by Gehrcken et al., published in 2025. The authors also addressed issues such as overactivation of T cells, suppression of NK cell function, release of immunosuppressive factors, and even promotion of tumor growth [125]. Facing these challenges, nanotechnology has been exploited to enhance pharmacokinetics, improving biodistribution and cytosolic delivery in recent years. Following, advances in STING agonist delivery methods were investigated for the past five years.

## 7. Recent Delivery Strategies for STING Agonists

Compared to free STING agonists, engineered delivery systems allow for lower dosing while achieving superior anti-tumor immune responses [60,128]. They provide enhanced serum stability and high specificity for tumor cells or immune cells within the TME, minimizing off-target effects, systemic inflammation, and treatment-related toxicity.

Dosta et al. developed an advanced nanoparticle (NP) formulation to enable a controlled release of STING agonist in the TME by covalently conjugating the CDNs to poly(beta)-amino-ester NPs via a cathepsin-sensitive linker. This linker is cleaved by endoproteases in the lysosomes, resulting in targeted release of the CDNs inside immune cells [129]. Alternatively, Su et al. designed pH-responsive polymeric nano-vaccines to co-deliver STING agonists and neoantigens [130]. To address the challenge of maintaining a therapeutically effective concentration of STING agonists within the TME, Lu et al. engineered polylactic-*co*-glycolic acid (PLGA) microparticles that remain at the side of injection and release encapsulated STING agonists long-term in a pulsative manner [131].

Another delivery strategy are liposomes; here, Chen et al. engineered STING-activating liposomal vesicles delivering a STING agonist as esterase-sensitive pro-drug for enhanced pharmacokinetic properties [132]. An especially relevant approach for targeting macrophages is based on NPs specifically designed to engage M2-polarized macrophages. Hussain et al. developed NPs composed of a peptide-expressing membrane functionalized with the peptides Pep20, matrix metalloproteinase-2 (MMP2), and M2-pep to achieve selective targeting of M2-like macrophages. These nanoparticles encapsulate 2′,3′-cGAMP, a potent STING agonist, to induce immunostimulatory effects within the TME [133].

A distinct approach utilizes the natural biological features of extracellular vesicles (EVs) to encapsulate and efficiently deliver STING agonists. Jang et al. purified extracellular vesicles from the supernatant of HEK293 cells and loaded them with STING agonists. These exoSTING EVs were tested on human macrophages and showed a preferential activation of M2-like macrophages [60].

Another strategy to extend the intratumoral release of STING agonists and to induce a sustained shift within the TME involved the use of hydrogels. Hydrogels are three-dimensional polymer matrixes that spontaneously self-assemble. Wang et al. conjugated the hydrophilic peptide moiety iRGD, that specifically targets cancer cells, to the hydrophobic cytostatic alkaloid camptothecin. This amphiphile conjugate self-assembled into supramolecular nanotubes. Via electrostatic complexation, the negatively charged STING agonist (c-di-AMP) condensed on the surface of the positively charged nanotubes. Under physiological conditions these nanotubes formed supramolecular hydrogels functioning as local reservoir of anti-cancer drugs [134]. These hydrogels could be applied during resection surgery. Delitto et al. embedded a STING agonist and a pancreatic cancer neoantigen into a hyaluronic acid hydrogel for implantation at the tumor site after incomplete resection surgery which reduced local tumor recurrence [135].

In conclusion, nanocarrier and conjugated STING agonists can overcome the challenges of first-generation agonists, but further optimization is still ongoing to improve targeting, bio distribution, and drug release mechanisms. Importantly, by selectively targeting both immune-suppressive myeloid cells and tumor cells, ADCs and nanocarrier-based STING agonists promote a localized pro-inflammatory shift within the TME.

## 8. Future Directions of STING Agonism in Cancer Therapy

The development of novel STING agonists, along with the advancement of innovative delivery strategies represents a rapidly evolving field of research. Emerging clinical data have highlighted both the potential and the limitations of STING as therapeutic target. In this context, further characterization of STING agonists regarding their ability to activate STING in macrophages and drive their polarization toward the M1-like subtype is highly anticipated. This could provide valuable insights into the role of key mediators in establishing a robust anti-tumor immune response, as well as the broader interplay of immune cells following STING activation. Such findings may support the development of more refined STING agonists that selectively target specific immune cell populations, thereby reducing STING-induced toxicity. Encouraging progress has been made in the development of targeted STING agonists which promise, in combination with advances in effective delivery systems, solutions to current challenges, potentially enabling a safer and more effective treatment across a broader spectrum of tumors.

The mechanism of STING agonists in combination with checkpoint inhibitors have been evaluated in various preclinical and clinical studies, revealing a synergistic effect that enhances anti-tumor responses [105,106]—even in tumors resistant to checkpoint blockade monotherapy [76,87,136]. Since STING activation often leads to checkpoint upregulation, combination therapy could further amplify the therapeutic effects of STING agonists [96,104]. In this context it remains a central challenge to identify optimal combination therapies of STING agonists with immune therapeutics and other anti-cancer drugs. The potential of STING agonists to transform cold, therapy-resistant tumors into hot, approachable ones [76,111], holds great promise to amplify the activity of additional therapeutic interventions. However, the heterogenicity of tumors and the variable genetic and immunologic background of patients limit our capacity to employ STING agonists under ideal settings. The assessment of patient biomarkers utilizing high throughput multidimensional screening and bioinformatical evaluation would open up opportunities to stratify patients and allow the design individualized treatment strategies comprising STING agonists as part of a tailored combination therapy.

In summary, the data reviewed in this article suggest that STING agonists, both alone and as combination therapy with immune checkpoint inhibitors, hold great promise for reprogramming the tumor microenvironment by shifting the suppressive immune compartment towards a pro-inflammatory state.

## Figures and Tables

**Figure 1 ijms-26-09008-f001:**
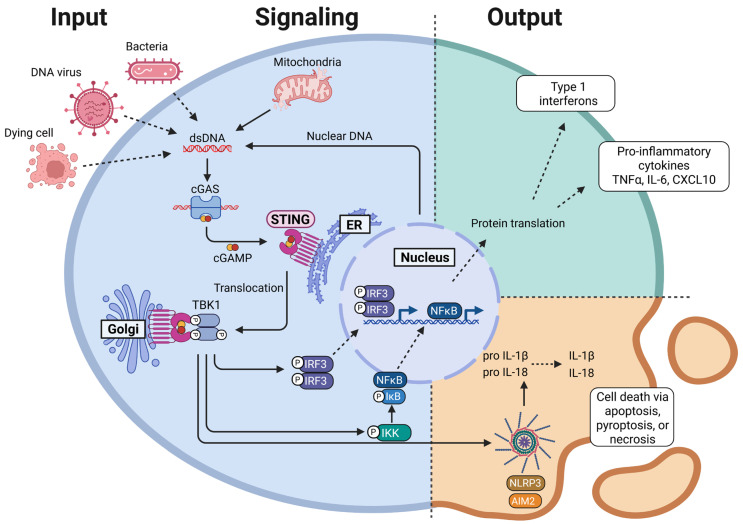
Overview of the STING signaling pathway in macrophages. Double stranded DNA (dsDNA) from endogenous or exogenous sources enters the cytoplasm and is detected by cyclic GMP-AMP synthase (cGAS). Upon activation, cGAS dimerizes and synthesizes the second messenger, cyclic GMP-AMP (cGAMP), which binds to endoplasmic reticulum (ER)-associated STING. STING undergoes dimerization and translocates to ER-Golgi intermediate compartments, where it recruits TANK binding kinase 1 (TBK1). TBK1 phosphorylates both STING and itself, creating a scaffold for the recruitment of downstream signaling molecules. TBK1 subsequently phosphorylates and activates IκB kinase (IKK), which in turn phosphorylates IκB, leading to the activation of the transcription factor nuclear factor kappa B (NF-κB). Additionally, TBK1 phosphorylates interferon regulatory factor 3 (IRF3), promoting its dimerization. Both IRF3 and NF-κB then translocate into the nucleus, where they induce the expression of pro-inflammatory cytokines and type 1 interferons (IFN-1). Furthermore, TBK1 can induce the activation of the NLRP3 (NOD-like receptor family, pyrin domain-containing-3) and AIM2 (absent in melanoma-2) inflammasomes. Created in BioRender. Binder, A.K. (2025) https://BioRender.com/t7f2r0a.

**Figure 2 ijms-26-09008-f002:**
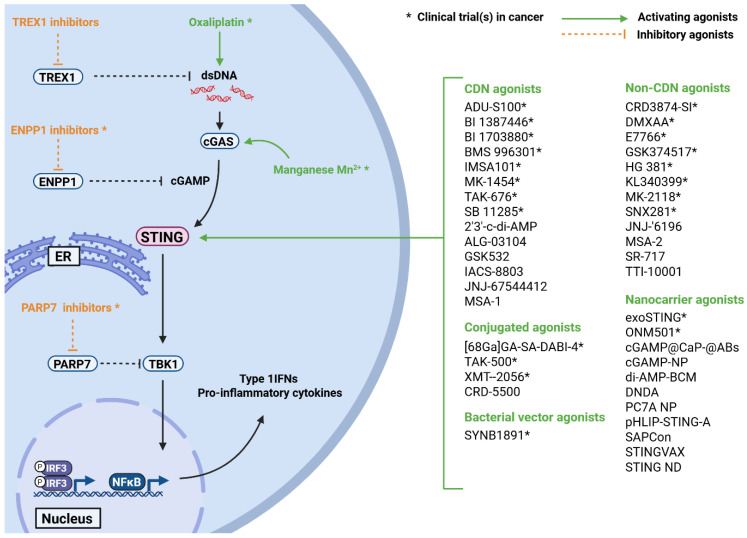
Overview of STING agonists in preclinical and clinical (*) evaluations as cancer treatment. Activating agonists are coded in green, inhibitory agonists in orange. Indirect STING activators like TREX1 inhibitors, Oxaliplatin, ENPP1 inhibitors, Manganese (Mn^2+^), and PARP7 inhibitors interfere with the STING signaling pathway by engaging with signaling molecules up- or downstream of STING. Direct STING agonists like cyclic dinucleotides (CDN), non-CDN, conjugated, bacterial vector, or nanocarrier agonists directly bind STING to enhance its signaling. Created in BioRender. Binder, A.K. (2025) https://BioRender.com/7y3180x.

**Table 1 ijms-26-09008-t001:** STING agonists in clinical trials for cancer therapy.

Agent ^1^ and Therapy		Route ^2^	Phase	SAE in Clinical Trials ^3^	Type of Cancer ^4^	NTC Code	Start	Status/Completion ^5^
**IMSA101 (GB492)**	Mono- therapy or +ICI/OC	IT	1/2a	Asthenia (5%), Acute respiratory failure (2.5%), Sepsis (2.5%), Pneumonia (2.5%), Angina pectoris (2.4%), Hepatorenal syndrome (2.4%)	Advanced solid tumors	NCT 04020185	23.09.2019	Completed 15.09.2023
	+PULSAR-ICI	IT	2		NSCLC or renal cell carcinoma	NCT 05846646	28.06.2023	Terminated 16.09.2024
	+PULSAR-ICI	IT	2		Solid tumors	NCT 05846659	07.07.2023	Terminated 20.11.2024
	+PULSAR	IT	2		Renal cell carcinoma	NCT 06601296	01.04.2025	Recruiting
	+ICI	IT	1		Advanced solid tumors	NCT 06026254	15.09.2023	Ongoing
**ADU-S100 (MIW 815)**	Mono- therapy or +Ipilimumab	IT	1	CRS (6.2%), Localized oedema (6.2%), Colitis (6.2%), Sepsis (6.2%), Acute kidney injury (6.2%), Pneumonia aspiration (6.2%)	Advanced/ metastatic solid tumors or lymphomas	NCT 02675439	28.04.2016	Terminated 06.08.2020
	+PDR001	IT	1		Solid tumors and lymphomas	NCT 03172936	08.09.2017	Terminated 18.12.2020
	+Pembrolizumab	IT	2		Head and neck cancers	NCT 03937141	28.08.2019	Terminated 10.06.2021
**SB 11285**	Mono- therapy or +Atezolizumab	IV	1	n/a	Advanced solid tumors	NCT 04096638	23.09.2019	Completed 16.07.2024
**TAK-676 (Dazo-stinag)**	Mono- therapy or +Carboplatin/5-FU/Paclitaxel	IT	Early 1	n/a	SCCHN	NCT 06062602	26.07.2021	Completed 15.11.2022
	+Pembrolizumab after Radiotherapy	IV	1		NSCLC, TNBC or SCCHN	NCT 04879849	09.09.2021	Completed 30.04.2024
	Mono- therapy or +Pembrolizumab	IV	1/2		Advanced/ metastatic solid tumors	NCT 04420884	22.07.2020	Recruiting
**E7766**	Mono- therapy	IT	1/1b	Upper gastrointestinal hemorrhage (4.2%), Vomiting (4.2%), Localized oedema (4.2%), Cerebral venous sinus thrombosis (4.2%), Hypertension (4.2%), Hypotension (4.2%)	Advanced solid tumors or lymphomas	NCT 04144140	24.02.2020	Terminated 26.07.2022
	Mono- therapy	Intra-vesical	1/1b		Non-muscle invasive bladder cancer	NCT 04109092	13.02.2020	Withdrawn 29.09.2022
**SYNB1891**	Mono- therapy or +Atezolizumab	IV, IT	1	CRS (15.6%), Sepsis (3.1%), Tracheal hemorrhage (3.1%), Transient ischaemic attack (3.1%), Hypoxia (3.1%), Pulmonary embolism (3.1%)	Advanced/ metastatic solid tumors or lymphomas	NCT 04167137	12.12.2019	Terminated 09.12.2021
**CDK-002 (exo-STING)**	Mono- therapy	IT	1/2	Grade 2 CRS (8.7%), Grade 1 pyrexia (4.4%)	Advanced/ metastatic solid tumors	NCT 04592484	15.09.2020	Completed 23.12.2022
**ONM-501**	Mono- therapy or +Cemiplimab	IT	1	n/a	Advanced solid tumors or lymphomas	NCT 06022029	13.10.2023	Recruiting

^1^ Agents: ICI = immune checkpoint inhibitor; OC = immune oncology drugs; PULSAR = personalized ultra-fractionated stereotactic adaptive radiotherapy. ^2^ Route of administration: IT = intratumoral; IV = intravenous. ^3^ Representation of most common serious adverse events (SAE) in clinical trials: CRS = cytokine release syndrome. ^4^ NSCLC = non-small-cell lung cancer; TNBC = triple-negative breast cancer; SCCHN = squamous-cell carcinoma of the head and neck. ^5^ Color coded according to status: Terminated/withdrawn: dark grey, completed: light gray, active: white.

**Table 2 ijms-26-09008-t002:** STING agonist mediated effects on cellular composition within the TME.

	CD8^+^ T Cells	NK Cells	M1-like	M2-like	DCs	T_H_1 Cells	B Cells	Tregs
**IMSA101 (GB492)**	↑	↑	↑	-	-	-	-	-
**ADU-S100 (MIW 815)**	↑	-	↑	↓	↑	-	-	-
**SB 11285**	↑	↑	↑	-	-	-	-	↓
**TAK-676 (Dazostinag)**	↑	↑	↑	-	-	-	-	-
**E7766**	↑	↑	↑	↓	↑	-	↑	-
**SYNB1891**	↑	-	-	-	-	-		-
**CDK-002 (exoSTING)**	-	-	↑	↓	-	↑	-	-
**ONM-501**	↑	↑	↑	-	-	-	-	-
**IACS-8803/** **IMGS-203**	↑	↑	↑	↓	↑	-	-	-
**MSA-2**	↑	↑	↑	-	-	-	-	-
**SR-717**	↑	↑	↑	-	-	-	-	-

↑ and green shading: upregulation; ↓ and red shading: downregulation.

**Table 3 ijms-26-09008-t003:** STING agonist mediated effects on soluble components within the TME.

	IFNα	IFNβ	IFNγ	TNFα	IL-6	IL-1β	IL-18	CXCL10	GM-CSF	TGFβ	IL-10
**IMSA101 (GB492)**	-	-	-	-	-	-	↑	-	-	↓	↓
**ADU-S100 (MIW 815)**		↑	↑		↑			↑			
**SB 11285**	↑	↑	-	↑	-	-	-	-	-	-	-
**TAK-676 (Dazostinag)**	↑	-	↑	↑	↑	-	-	↑	-	-	-
**E7766**	-	↑	-	↑	↑	-	-	↑	-	-	-
**SYNB1891**	↑	↑	↑	↑	↑	↑	-	-	↑	-	-
**CDK-002 (exoSTING)**	-	↑	-	-	-	-	-	↑	-	-	-
**ONM-501**	-	↑	-	-	-	-	-	↑	-	-	-
**IACS-8803/** **IMGS-203**	-	-	-	-	↑	-	-	-	-	↓	-
**MSA-2**	-	↑	-	↑	↑	-	-	↑	-	-	-
**SR-717**	-	↑	-	↑	↑	-	-	↑	-	-	-

↑ and green shading: upregulation; ↓ and red shading: downregulation.

## Data Availability

Not applicable.

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
