# Peer review of "STINGing Cancer: Development, Clinical Application, and Targeted Delivery of STING Agonists"

_ijms, 2025, doi:10.3390/ijms26189008_

Round 1
Reviewer 1 Report
Comments and Suggestions for Authors
This review by Nerdinger et. al. discusses the current state of clinical trials employing STING agonists for anticancer therapy, and the optimized delivery systems for greater potency combined with nanotechnologies to improve efficacy. It is well written and organized, and the background section sufficiently prepares the reader for the discussion of the current clinical trials. It is expected that many readers will find this review very helpful and the detailed summary of clinical trial results will be impactful. There are two minor errors that must be corrected:
Line 95 has a reference error that must be fixed.
Line 504 “side” should be “site”
The only major change that I recommend is the discussion and characterization throughout the paper of M1 and M2 macrophages. The problem with the M1/M2 macrophage nomenclature is that it is an oversimplified dichotomy for a complex and dynamic cellular system, as macrophages display plasticity and exist on a spectrum of functional states rather than in two distinct types. The terms often lead to inconsistent and confusing use of markers in different experimental settings, misrepresenting macrophage biology and hindering research and therapeutic development. The M1/M2 paradigm has been a useful tool in characterizing and investigating macrophage responses when their complex biology began to be scrutinized, but now as the field is maturing it is outdated and incorrect. While somewhat observable in inbred mice used in the laboratory, the M1/M2 paradigm is especially flawed when applied to humans where macrophages clearly display a spectrum of responses, frequently having high levels of both M1-like and M2-like cytokines in the same cell. Even though many groups still operate and publish using this paradigm, it should no longer be considered a viable framework to operate under. Therefore, this review needs to include a paragraph that adequately discusses this issue. At the very least, macrophages should be referred to as M1-like or M2-like. What seems to be more accepted is to refer to them as inflammatory and non-inflammatory macrophages, although M2 macrophages clearly have an inflammatory component. Neither schema is perfect, but certainly better than M1/M2.
Below are a few recent articles supporting this issue that would be useful for starting a deeper dive into this topic if needed:
https://pmc.ncbi.nlm.nih.gov/articles/PMC4965179/
https://pmc.ncbi.nlm.nih.gov/articles/PMC10407193/
https://www.google.com/url?sa=t&source=web&rct=j&opi=89978449&url=https://www.sciencedirect.com/science/article/pii/S0896627322009539&ved=2ahUKEwip8u--3KaPAxX2pIkEHcNtCrYQFnoECFIQAQ&usg=AOvVaw2V6X6MinIh81-4abJDjRkK
Author Response
This review by Nerdinger et. al. discusses the current state of clinical trials employing STING agonists for anticancer therapy, and the optimized delivery systems for greater potency combined with nanotechnologies to improve efficacy. It is well written and organized, and the background section sufficiently prepares the reader for the discussion of the current clinical trials. It is expected that many readers will find this review very helpful and the detailed summary of clinical trial results will be impactful. There are two minor errors that must be corrected:
Answer: We want to thank the reviewer for this positive evaluation and for the helpful comments. Please find our answers below
Line 95 has a reference error that must be fixed.
Answer: This has already been done by the editorial office.
Line 504 “side” should be “site”
Answer: corrected
The only major change that I recommend is the discussion and characterization throughout the paper of M1 and M2 macrophages. The problem with the M1/M2 macrophage nomenclature is that it is an oversimplified dichotomy for a complex and dynamic cellular system, as macrophages display plasticity and exist on a spectrum of functional states rather than in two distinct types. The terms often lead to inconsistent and confusing use of markers in different experimental settings, misrepresenting macrophage biology and hindering research and therapeutic development. The M1/M2 paradigm has been a useful tool in characterizing and investigating macrophage responses when their complex biology began to be scrutinized, but now as the field is maturing it is outdated and incorrect. While somewhat observable in inbred mice used in the laboratory, the M1/M2 paradigm is especially flawed when applied to humans where macrophages clearly display a spectrum of responses, frequently having high levels of both M1-like and M2-like cytokines in the same cell. Even though many groups still operate and publish using this paradigm, it should no longer be considered a viable framework to operate under. Therefore, this review needs to include a paragraph that adequately discusses this issue. At the very least, macrophages should be referred to as M1-like or M2-like. What seems to be more accepted is to refer to them as inflammatory and non-inflammatory macrophages, although M2 macrophages clearly have an inflammatory component. Neither schema is perfect, but certainly better than M1/M2.
Below are a few recent articles supporting this issue that would be useful for starting a deeper dive into this topic if needed:
https://pmc.ncbi.nlm.nih.gov/articles/PMC4965179/
https://pmc.ncbi.nlm.nih.gov/articles/PMC10407193/
https://www.google.com/url?sa=t&source=web&rct=j&opi=89978449&url=https://www.sciencedirect.com/science/article/pii/S0896627322009539&ved=2ahUKEwip8u--3KaPAxX2pIkEHcNtCrYQFnoECFIQAQ&usg=AOvVaw2V6X6MinIh81-4abJDjRkK
Answer: This is indeed a critical issue. We completely agree that the black and white classification of macrophages into M1 and M2 is outdated and inaccurate. Nevertheless, many of the resources we cite in our manuscript adhere to this nomenclature. We have included a detailed description of this problem quite at the beginning of our manuscript, (line 66 to 89) and we have changed the terms M1 and M2 to M1-like and M2-like throughout the manuscript to indicate that they are not accurate terms.
Reviewer 2 Report
Comments and Suggestions for Authors
The review is well written and comprehensive. The review is timely as it summarizes the clinical trial outcomes of STING agonists at a time when there are remarkable advances.
Comments :
Figure 1 : Line 95 mentions " error! reference source not found - please include the reference
Would it be possible to improve the readability of figure 1 ? Perhaps by using color coding to distinguish components. The abbreviations are better expanded when they first occur in the text instead of the figure legend
Figure 2 : It is also very dense - perhaps separating the direct and indirect agonists would be helpful either through color coding . The legend is also long - simplifying this would be helpful.
Table 1 is an incredible resource . At first glance it is hard to identify clinical trial status - perhaps separating them into current vs completed would be helpful.
Including some insight into why some of the past clinical trials failed would be helpful. Additionally including some section about biomarkers would be helpful .
Expanding the limitations section to focus more on the patient response as well as future directions would make this more helpful. Talking about personalized medicine and how that could help STING agonists patient response ?
Comments on the Quality of English Language
The overall article is well written but could benefit from some minor editing.
For example there is a line that reads " in regards of" - perhaps rewriting this in a more conventional way such as with regards to would improve the readability
-Some of the sections do feel disconnected.
Author Response
Comments and Suggestions from Reviewer 2
The review is well written and comprehensive. The review is timely as it summarizes the clinical trial outcomes of STING agonists at a time when there are remarkable advances.
Answer: We want to thank the reviewer for this positive evaluation and for the helpful comments. Please find our answers below
Comments :
Figure 1 : Line 95 mentions " error! reference source not found - please include the reference
Answer: This mistake has already been corrected by the editorial office.
Would it be possible to improve the readability of figure 1 ? Perhaps by using color coding to distinguish components. The abbreviations are better expanded when they first occur in the text instead of the figure legend
Answer: We have adjusted the colors to depict the activation substances in red/orange and made some additional modifications. We have also moved the figure down, so the abbreviations are defined in the main text above the figure. We still would like to keep the definitions in the legend as well, to avoid that readers have to search through the text, while looking at the figure. Only the definition of STING itself was removed.
Figure 2 : It is also very dense - perhaps separating the direct and indirect agonists would be helpful either through color coding . The legend is also long - simplifying this would be helpful.
Answer: We have color-coded inhibitors that indirectly activate STING vs agonistic activators and removed the redundant definition of STING from the legend
Table 1 is an incredible resource . At first glance it is hard to identify clinical trial status - perhaps separating them into current vs completed would be helpful.
Answer: We originally also thought about splitting the table accordingly. However, then trials with the same substance would be separated. We now decided to color-code the trials as three categories: Competed vs terminated/withdrawn vs active to make the table more intelligible
Including some insight into why some of the past clinical trials failed would be helpful. Additionally including some section about biomarkers would be helpful .
Answer: We completely agree, but unfortunately, the available information is scarce. Wherever we found respective information, we included it already in the text.
Expanding the limitations section to focus more on the patient response as well as future directions would make this more helpful. Talking about personalized medicine and how that could help STING agonists patient response ?
Answer: Since a good review specially on this topic has recently been published elsewhere (Gehrcken et al), we did not want to include too much redundant information. However, we now indicate more clearly, what this other review is about. (line 577-580). We also now elaborate more on the future perspectives to use STING agonists in concert with other therapeutic interventions in a personalized manner, aided by bioinformatic analyses.(line 653-664)
Comments on the Quality of English Language
The overall article is well written but could benefit from some minor editing.
For example there is a line that reads " in regards of" - perhaps rewriting this in a more conventional way such as with regards to would improve the readability
-Some of the sections do feel disconnected.
Answer: The respective sentence was rephrased and the manuscript was proofread again.